


# Deformation characteristics and exploratory data analysis of rainfall-induced
rotational landslide: A case study of the Zhutoushan landslide in Nanjing,
China
Weiguo Li[1,2], Yali Liu[2], Libing Yang[2], Yanhong Chen[2]
[1] Department of Geoinformatics, VŠB – Technical University of Ostrava, 708 00, Ostrava, Czech Republic
[2] College of Land Science and Planning, Hebei GEO University, China
**Abstract-** Due to the complex geological structure of landslides, the installation of a monitoring network could be useful for a variety
of scopes studying the possible evolution of a landslide for early warning, and the occurrence of disasters of different types landslides is
different not only in the form of deformation, but also in the trigger factor. In the process of landslide monitoring, due to equipment
failure and external factors, data loss or abnormal are inevitable. In this paper, through the processing and analysis of the monitoring
data of the Zhutoushan landslide, the landslide is rotational landslide which is caused by the rainfall. The box plot is used to detect
outliers, and the polynomial fitting function and the moving average denoise method are compared to repair the data, and the latter is
better. Through the exploratory analysis of GNSS data, the correlation between monitoring points at different locations is found, which
provides a basis for the identification of landslide types.

**Keyword**: rotational landslide, trigger factor, exploratory data analysis
## 1. Introduction
Landslide is a kind of common geological hazard, which occurs all over the world and seriously threatens the safety
of life and property (Calcaterra et al., 2012; Yen-Yu et al., 2019; Mustafa et al., 2015), rainfall is a recognized trigger
(Monsieurs et al., 2019. Sidle and Bogaard, 2016). Many authors have carried out relevant studies and proposed
rainfall threshold and established corresponding models to predict the occurrence of landslides (Bappaditya et al.,
2019; Elise et al., 2019), and have developed the territorial early warning systems for rainfall induced landslide (Luca
et al., 2018). Some authors have also suggested that rainfall information is not sufficient to predict the occurrence of
landslides because it does not reflect soil moisture conditions (Koizumi et al., 2019). Only one monitoring method is
not enough to accurately monitor the landslide deformation. Currently, there are hydrological monitoring, geological
monitoring and surface monitoring. It is reasonable to set alarm thresholds for multiple parameters (Pecoraro et al.,
2019). However, for some landslides, the rainfall will cause changes in other monitoring parameters.
For the landslide early warning system, the method of mathematical model is often used to predict, and good results
have been achieved (Fasheng et al., 2018; Xing et al., 2018). However, the precondition of establishing mathematical
model is to ensure the integrity and validity of monitoring data. In the process of landslide monitoring, the loss or
abnormality of monitoring data caused by monitoring equipment failure or external factors is inevitable (Yong et al.,
2019). For abnormal data, it is necessary to know clearly whether it is caused by disturbance or equipment failure or
landslide deformation, so as to avoid triggering false alarm.
Due to the complex geological structure of landslides, the deformation of monitoring points at different locations
is closely related to the geological features of specific locations (Yong et al., 2019). This paper provides insight into
landslide type and gives the relationship between rainfall and other monitoring data through the analysis of the
monitoring data of the Zhutoushan landslide in China and how to judge which data is outlier through exploratory data
analysis.
## 2. Study area
The Zhutoushan landslide lies above the residential area of Yongning town, Pukou district, Nanjing city, Jiangshu
province, China. The center of area is located at    118°39′37″  east longitude and 32°09′24″  north latitude(Fig 1).
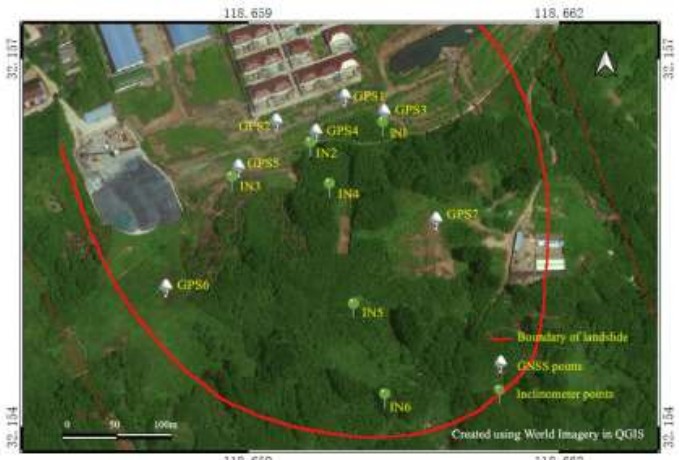

Fig 1. Location of the studied landslide site
The working area is dissected by numerous faults linking the Zhutou mountain fault zone. The geology is composed
of heavily deformed sandstone, siltstone, marlstone, limestone and soil. In the 1970s, there were large-scale mining
activities at the foot of the Zhutoushan. In the process of land management, unreasonable excavation leads to many
landslide disasters, and many houses are destroyed at the foot of slope, resulting in large property losses and a large
number of people threatened by landslides.
## 3. Material and methodology
In order to monitor the deformation of zhutoushan landslide in real time, the automatic deformation monitoring and
warning system based on GNSS is adopted. The system integrates GNSS high-precision positioning technology,
wireless communication technology, database technology, General Packet Radio Service (GPRS) communication
technology, sensor technology and other new technology achievements, and can monitor the landslide in real time
and timely predict and analyze the monitoring results.
According to the design requirements and field investigation, This system was composed of one GNSS reference
station which was located outside of the landslide, eight GNSS monitoring stations (Fig 1)(the GPS8 is outside the
area affected by landslide deformation), six inclinometer monitoring points and each point was installed with four
sensors at each depth to detect slope deformation, four water level monitoring points, three pore water pressure(PWP)
monitoring points, one rainfall monitoring point which was installed at the edge of the landslide(Fig 2 and Fig 3),
one soil water content(SWC) monitoring point and two video monitoring points. The system was initiated in July
2017, and data were sent to the computer center in real time using the wireless sensor network technology.

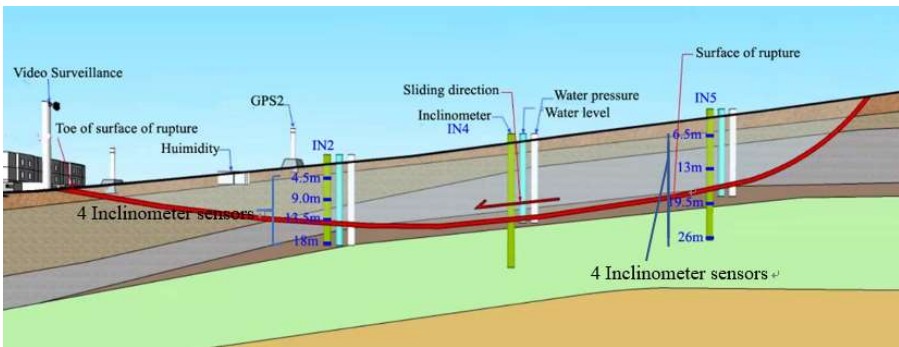




Fig 2. Positions of the monitoring instruments

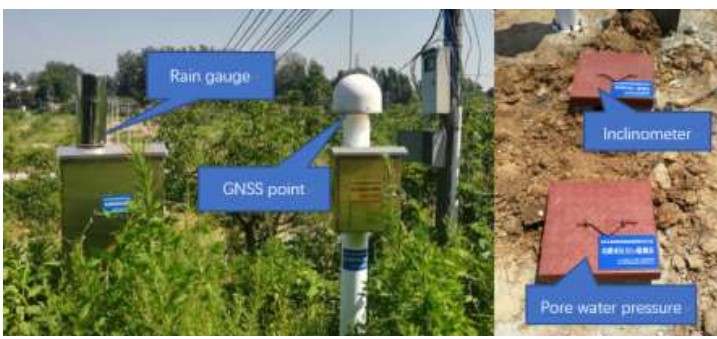


Fig 3. Monitoring instruments on the landslide

The methodology used involved: (a) GNSS data, inclinometer data, rainfall data, soil water content data and water
pressure data come from the landslide early warning system, (b) data analysis using Exploring Data Analysis (EDA)
method to establish the relationship between rainfall and the other data, and find outlier and characteristics of GNSS
data.

## 72   4. Analysis and results

### 73   4.1 Rainfall and Displacement

In most of the cases, the main trigger of landslides is heavy or prolonged rainfall. A detailed review of the literature
reveals that numerous landslides have been related to rainfall (Heyerdahl et al. 2003; Glade et al. 2000; Zezere et al.
2005). During heavy rains, water seeps into the ground and travels through unsaturated soils. This water may perch
on lower permeability materials or a drainage barrier such as bedrock and highly impermeable clays.
The rainfall has a great impact on the displacement of GPS surface and underground inclination. On August 15,
2018, less than 2 mm of rainfall caused changes in the horizontal displacement of the surface, but had little effect on
the changes in the inclination and elevation. The rainfall over the three days of December 25, 26 and 27 in 2018 was
178mm, 406mm and 313mm, respectively. That caused dramatic changes in horizontal displacement, vertical
displacement and inclination (Fig 4). The displacement of different depth for inclinometer, affected by rainfall, are
also different. The deformation of the surface and buried depth of 4.5m exceeded those of the buried depth of 9m and
13.5m.

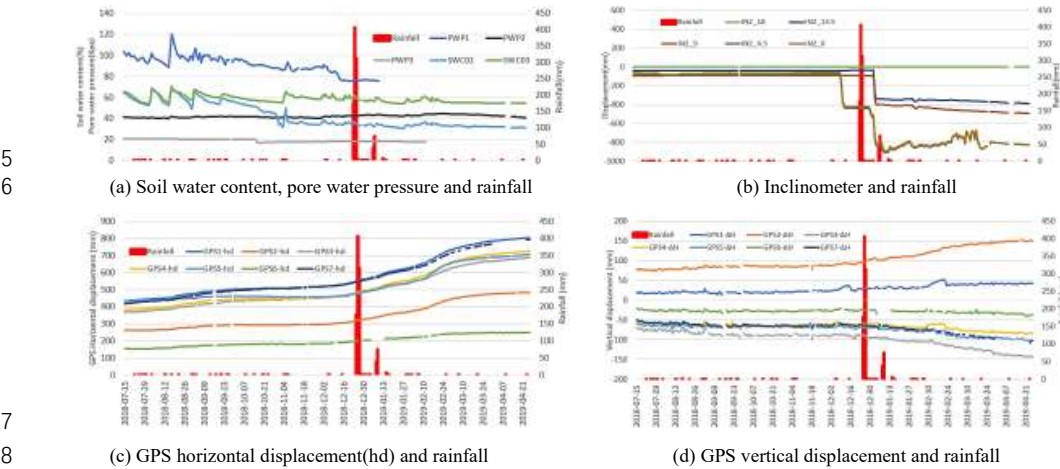


(a) Soil water content, pore water pressure and rainfall            (b) Inclinometer and rainfall

(c) GPS horizontal displacement(hd) and rainfall         (d) GPS vertical displacement and rainfall

Fig 4. Relationship between rainfall and the other monitoring data


Rainfall has little effect on pore water pressure. The influence of rainfall on soil water content is also relatively
small, only affecting its fluctuation in a small range (Fig 4a). Even the rainfall on December 26, 2018 was 406 mm,
these changes are not obvious.
**4.2 Detect outliers in raw data**
In the process of GPS data collection and transmission, measurement error and random noise are inevitable. By
establishing the corresponding mathematical model or error processing, the influence of error and noise on the
original data can be reduced. However, for outlier, it greatly affects the quality of data and the judgment and modeling
of original data. Therefore, in establishing the corresponding mathematical model, it is necessary to judge the outlier
of the original data and remove the outliers.
Box plots can be used to detect outliers in raw data. GPS1 has an abnormal value in the elevation direction, and x,
y, and horizontal directions are normal (Fig 5). The vertical displacement of GPS1 can be fitted by a basic 20 days
moving average method. With this method, each observation is replaced by an average (Fig 6). But some important
information can be covered. Moving average denoising method is suitable (LI et al., 2016; JI et al., 2015; JI et al.,
2015 May). Using this method, the Root Mean Square Error (RMSE) is used to judge the outliers and replace them
with the average value (Fig 7). The other values are still observations. Because some important information from the
raw data are available.
For the same data, polynomial fitting model is adopted, and it is found that the correlation coefficient of the second
method is better than that of the first method. That is to say, the accuracy of the polynomial fitting model is improved
after the outliers are removed. However, there will be a problem. How to judge whether the abnormal value is caused
by the measurement error, or whether it is a real deformation value, and whether an alarm is required. This requires
comprehensive consideration of various factors to make a comprehensive judgment. Since the operation of the system,
outliers have been caused by equipment fault.

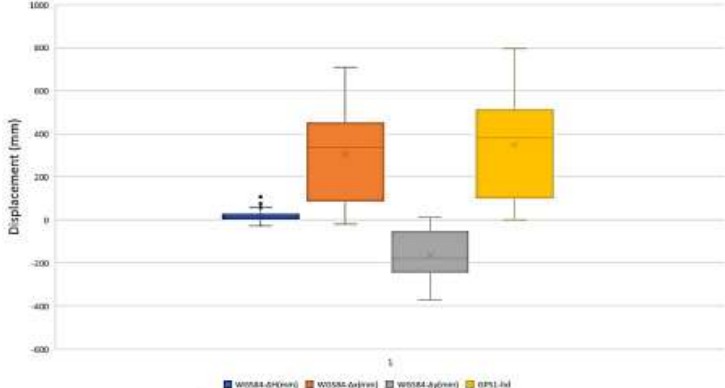

Fig 5. Box plots of GPS


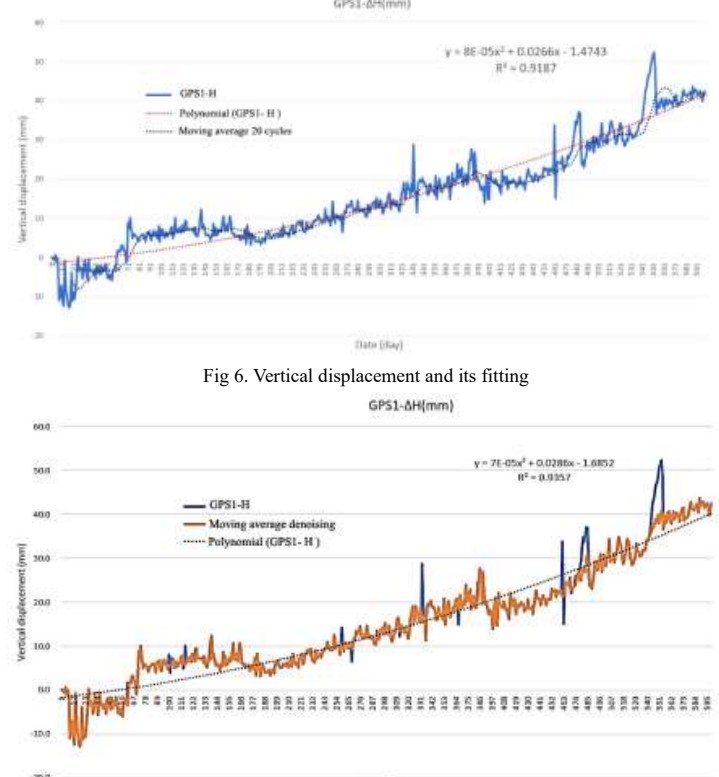

Fig 6. Vertical displacement and its fitting
Fig 7. Vertical displacement and its fitting
**4.3 Exploratory data analysis (EDA)**
Exploratory data analysis (EDA) is an approach to analyzing data sets to summarize their main characteristics, often
with visual methods, and this method have been successfully applied to a variety of issues (Bondarev 2019).
It can be seen from the scatter plots that R squared is greater than or equal to 0.9 (Fig 8 and 10). R squared of GPS3
is better than the others, its value is 0.970, and R squared of GPS1 is the lowest, some points are far away from the
line. The variables on both axes are rescaled to standard deviational units, so any observations beyond the value of 2
can be designated as outliers (Fig 9a) (Anselin, 2005). When getting started with brushing in the scatter plot, the
regression line is recalculated on the fly, reflecting the slope for the data set without the current selection (Fig 9b), R
squared of GPS1 will increase to 0.911.

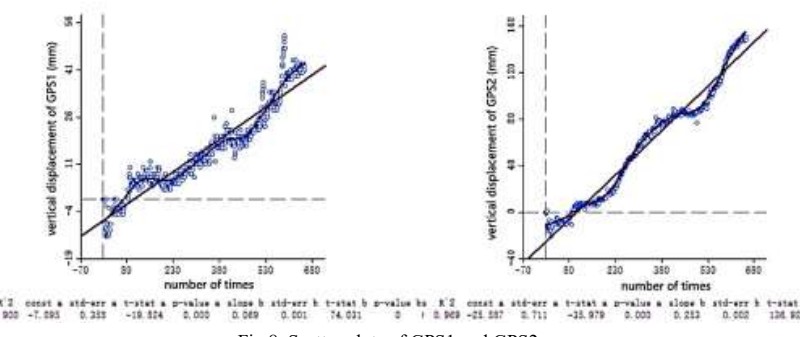

Fig 8. Scatter plots of GPS1 and GPS2

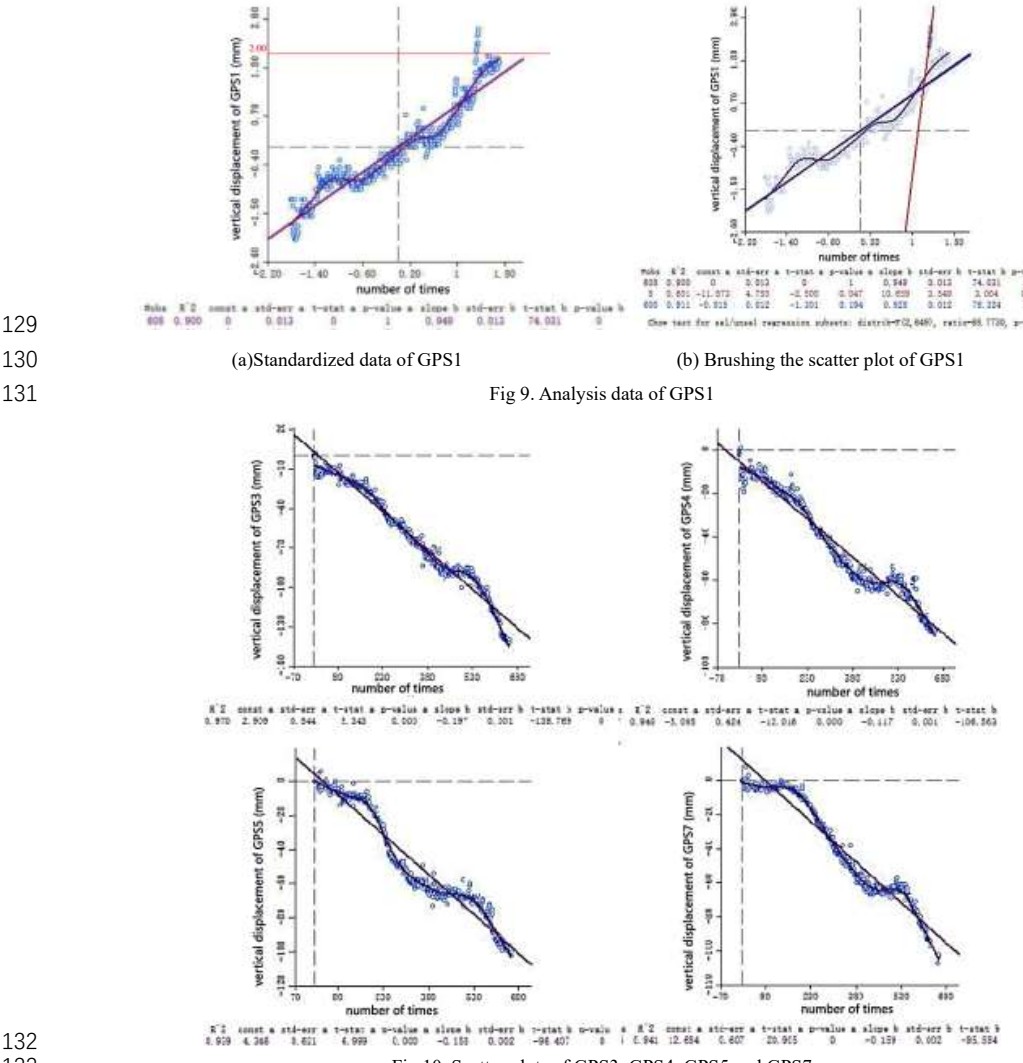


(a)Standardized data of GPS1                (b) Brushing the scatter plot of GPS1

Fig 9. Analysis data of GPS1

Fig 10. Scatter plots of GPS3, GPS4, GPS5 and GPS7

Because vertical displacements of GPS1 and GPS2 are up, the others are down, the trend of vertical deformation
is up or down, and the relationship between them is positive correlation. If one trend of vertical deformation is up
and one trend of vertical deformation is down, their relationship is negative correlation (Fig 11).



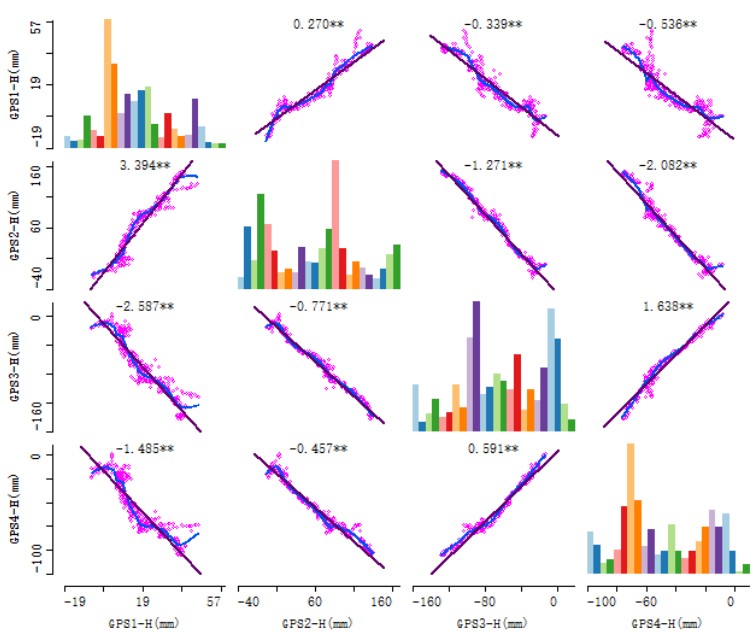


Fig 11. Scatter plot matrix between GPS1,GPS2,GPS3 and GPS4

**4.4 Landslide surface displacement**
The horizontal displacement direction of the eight GPS stations reflects the sliding tendency of the landslide body in
the horizontal direction. It can be seen from the figure that the landslide as a whole in the direction of the northwest,
and the azimuth angle is about from 310° to 330° (Fig 12). The largest horizontal displacement is GPS1, GPS8
horizontal displacement is the smallest, and the value is less than 50mm, indicating that the point is currently stable.

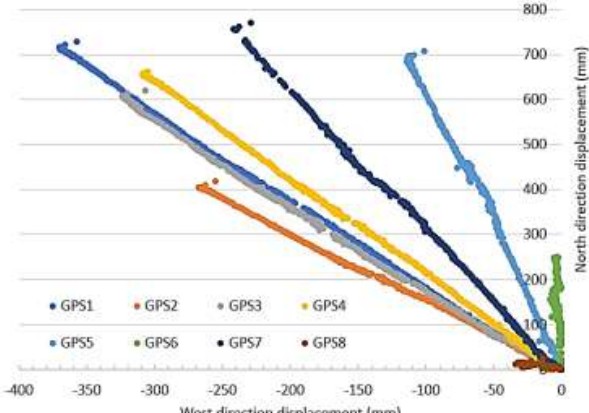


Fig 12. GPS displacement from July 14th, 2017 to May 1st, 2019


The deformation of the landslide body is not a change in one direction, but a change in three directions (Fig 13).

The deformation of the north is greater than the deformation of the west, so the direction of landslide displacement
is transformed into the north. The GPS1 and GPS3 monitoring points are rising in the vertical displacement, the others
are opposite.

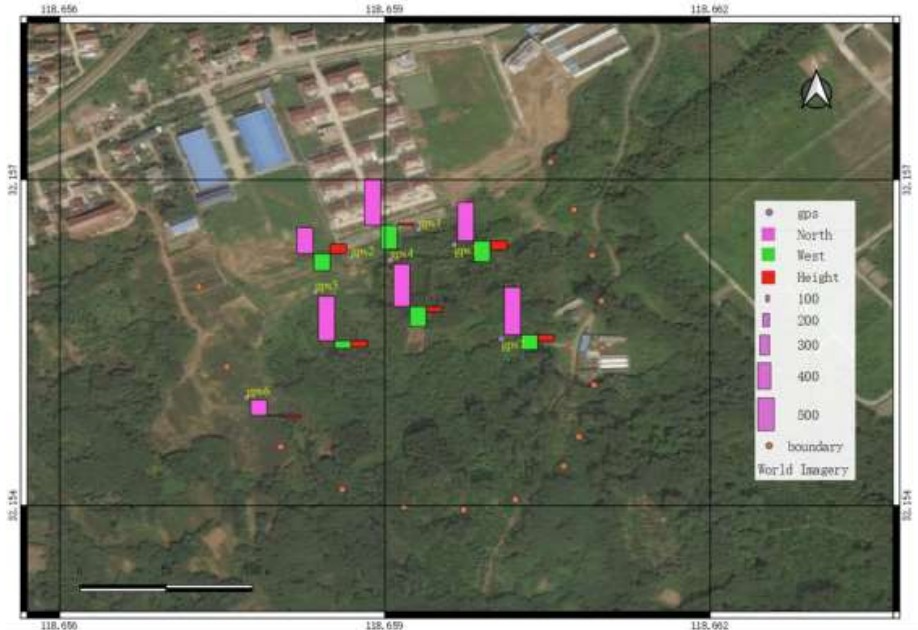

Figure 13. Surface displacement visualization (2019-04-08)

The monitoring time of the Zhutoushan landslide is from July 14, 2017 to April 8, 2019. From the data analysis of
GNSS monitoring points, except for GPS6 and GPS8, the vertical displacement of other points is larger (Fig 14). The
horizontal displacement of the GPS1 point is the largest, reaching 792 mm. From the perspective of vertical
displacement, GPS1 and GPS2 points go up, other points go down, and GPS2 and GPS3 are larger than others, the
largest is GPS2 point, reaching 149.8mm. From the perspective of deformation rate, the average rate of 8 points is
2mm per day, indicating that the landslide is in a stable state as a whole, but observation should be strengthened,
especially for GPS1 and GPS2 at the low of the landslide. Therefore, large deformation, as time goes on, the
possibility of sudden deformation of the lower part of the landslide body will increase, causing the entire landslide
body to collapse.

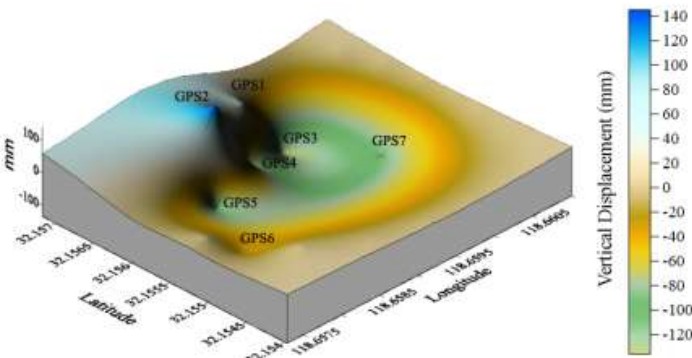

Fig 14. 3D graph on April 8th. 2019


**5. Discussion**





167 According to classification of landslides (Cruden and Varnes 1996), zhutoushan is rotational landslide (Fig 15).

168 Squeezed by the upper landslide, GPS1 and GPS2 that lie in the toe of surface of rupture rise in vertical direction.

169 Other GPS monitoring points slip down under the influence of gravity.

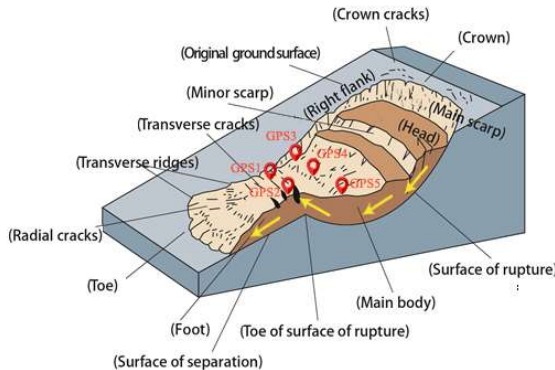


171        Fig 15. Zhutoushan landslide type (based on Varnes 1978)

172  It is very dangerous to ignore the existence of outliers in the raw data. If the outliers are included in the process of

173 data processing and analysis with exclusion, it will have a negative impact on the results. The box plots are just a

174 great tool for detecting outliers from the raw data. Polynomial fitting models and moving average noise reduction

175 methods can be used to repair outliers. From the data analysis point of view, the accuracy of the latter is better than

176 the former. The polynomial fitting model is a kind of mathematical model which can be fitted by all the raw data

177 including outliers, it will remove some important information from the raw data, and the moving average denoise

178 method will retain some. According to the setting step, the precision and retained information will also be different,

179 this requires setting the corresponding step size according to the specific project. In addition, it is necessary to

180 emphasize the judgment of the abnormal value. Whether the outlier is caused by other external factors or due to

181 landslide deformation. This requires a comprehensive judgment to avoid misjudgment and threat the people's lives

182 and damage to property. If the outliers are caused by the deformation of the landslide and exceed the deformation

183 warning value, the system should send an alarm to remind people to pay attention to safety. Otherwise, the outliers

184 can be removed from the raw data.

185  Due to the complex geological structure of the landslide, the deformation of the monitoring points at different

186 locations is related to the geological features of the landslide body and the type of landslide. Through exploratory

187 analysis of surface GNSS data, the relationship between different monitoring points is positive correlation and

188 negative correlation, which is consistent with most of the same type of landslide deformation. After standardizing the

189 data, the outliers can also be detected to improve the quality of the data.

## 6. Conclusions

191 In this paper, through the processing and analysis of the monitoring data of the Zhutoushan landslide, the landslide

192 is rotational landslide which is caused by the rainfall. The box plot is used to detect outliers, and the polynomial

193 fitting function and the moving average denoise method are compared to repair the data, and the latter is better.

194 Through the exploratory analysis of GNSS data, the correlation between monitoring points at different locations is

195 found, which provides a basis for the identification of landslide types.

196  In addition, multiple monitoring methods can be used to enhance the monitoring of the landslide, such as

197 meteorological monitoring and geological monitoring, and the mutual verification of the landslide deformation can

198 also be performed between multiple monitoring means.


With the development of landslide monitoring equipment, data collection, transmission and storage technology, it
is one of the development directions of landslide monitoring information processing in the future to mine the complex
relationship between massive monitoring data and various monitoring data.
## Software
All data processing and spatial analysis were performed by QGIS 3.6.1、Surfer 15 、Geoda and Matlab R2016b
software.
## Acknowledgement
The authors want to thank Jiangsu Kebo Space Information Technology Limited Company for kindly providing all
the data of the landslide. Moreover, they want to thank the reviewers for the pertinent suggestions that improved the
final version of the manuscript.
The author thank Ing. Kačmařík Michal, Ph.D. for his critical comments and suggestions, which greatly improved
the quality of our manuscript and map.

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
