# Peer review of "Deformation characteristics and exploratory data analysis of rainfall-induced rotational landslide: A case study of the Zhutoushan landslide in Nanjing,"

_Natural Hazards and Earth System Sciences, 2020_

## Referee Comment (RC1) · Anonymous Referee #1 · 4 Jul 2020

1. The novelty is not clear. Please improve this part in "Introduction" section. 2. Recommend the limitations of the research methods can be discussed in your article.

---

## Author Comment (AC1) · 9 Jul 2020

Thanking you for your suggestion, which prompted us to improve and enhance our manuscript. Exploratory data analysis (EDA) is an approach to analyzing data sets to summarize their main characteristics, often with visual methods, and plays a major role in obtaining insights from data (Aindrila et al., 2018). This method have been successfully applied to a variety of issues (Bondarev 2019), such as Computer Graphics (Endert et al., 2017), Bioinformatics (Pabinger et al., 2014 and Dunn et al., 2016), Meteorology (Rautenhaus et al., 2017), Traffic (Nikolaos et al.,2020), Crops (Peng et

al., 2013) and so on. For landslide, Muhammad Qarinur 2015 focus on determine the correlation between landslide run out distance against high, slope, and volume based on mechanisms and causes of soil or rock mass movement by using EDA tool. So EDA were implemented, for the first time, to describe the relation between the location of GPS monitoring points and the trend of their displacements, detected the outliers from the raw data and verified the deformation characteristics of rotational landslide in terms of 3D graph. So this is our novelty that we want to say. We will improve this part in "Introduction" section in our new manuscript.

Reference Aindrila Ghosh, Mona Nashaat, James Miller, Shaikh Quader, Chad Marston. A comprehensive review of tools for exploratory analysis of tabular industrial datasets. Visual Informatics 2 (2018) 235–253. https://doi.org/10.1016/j.visinf.2018.12.004 Dunn, Jr., W., Burgun, A., Krebs, M.-O., Rance, B., 2016. Exploring and visualizing multidimensionaldataintranslationalresearchplatforms.Brief.Bioinform.18 (6),1044–1056.http://dx.doi.org/10.1093/bib/bbw080. Endert, A., Ribarsky, W., Turkay, C., Wong, B.W., Nabney, I., Blanco, I.D., Rossi, F., 2017.Thestateoftheartinintegratingmachinelearningintovisualanalytics. In: Computer Graphics Forum (36). Wiley Online Library, pp. 458–486. http: //dx.doi.org/10.1111/cgf.13092. M.P. Jácome-Paz,I.A. González-Romo,R.M. Prol-Ledesma,M.A. Torres Vera,D. Pérez-Zárate,A.A. Rodríguez-Díaz,A.M. Estrada-Murillo. Multivariate analysis of CO 2 , H 2 S and CH 4 diffuse degassing and correlation with fault systems in Agua Caliente - Tzitzio, Michoacán, México[J]. Journal of Volcanology and Geothermal Research,2020,394. DOI:10.1016/j.jvolgeores.2020.106808 Muhammad Qarinur. LANDSLIDE RUNOUT DISTANCE PREDICTION BASED ON MECHANISM AND CAUSE OF SOIL OR ROCK MASS MOVEMENT. Journal of the Civil Engineering Forum. 2015,1(1) N. V. Bondarev (2019). Classification and Prediction of Sodium and Potassium Coronates Stability in Aqueous-Organic Media by Exploratory Data Analysis Methods. Russian Journal of General Chemistry, 2019,89(2): 281–291. DOI: 10.1134/S1070363219020191 Nikolaos-Fivos Galatoulas, Konstantinos N. Genikomsakis, Christos S. Ioakimidis.

Spatio-Temporal Trends of E-Bike Sharing System Deployment: A Review in Europe, North America and Asia. 2020, 12(11). DOI:10.3390/su12114611 Pabinger, S., Dander, A., Fischer, M., Snajder, R., Sperk, M., Efremova, M., Krabichler, B., Speicher, M.R., Zschocke, J., Trajanoski, Z., 2014. A survey of tools for variant analysis of next-generation genome sequencing data. Brief. Bioinform. 15(2),256–278.http://dx.doi.org/10.1093/bib/bbs086. Peng Hongxing, Zou Xiangjun, Guo Aixia1 Xiong Juntao, Chen Yan. Color Model Analysis and ïijšecognition for Parts of Citrus Based on Exploratory Data Analysis. Transactions of the Chinese Society for Agricultural Machinery. 2013,44(S1):253-259. doi:10.6041/j.issn.1000-1298.2013.S1.045 Rautenhaus, M., Böttinger, M., Siemen, S., Hoffman, R., Kirby, R.M., Mirzargar, M., Röber, N., Westermann, R., 2017. Visualization in meteorology-a survey of techniques and tools for data analysis tasks. IEEE Trans. Vis. Comput. Graphics early-view,http://dx.doi.org/10.1109/TVCG.2017.2779501.

---

## Author Comment (AC2) · 10 Jul 2020

The authors are grateful to the reviewer, who offered the constructive suggestions to enhance our manuscript. EDA can only detect outliers in a dataset and can not identify the cause of the outliers which is caused by data transmission or deformation of the landslide. Simply removing outliers can add further bias to the analysis, so EDA need to enable the user's engagement in the terms of user's feedback. This is our main challenges and limitations for exploring the dataset. For GNSS data in landslide monitoring which is a single variable data, we are mainly analyzed by using box plots and scatter

plots. In order to monitor deformation of landslide, multi-variable data can be collected by using a variety of monitoring methods and means. The EDA is usually difficult to describe the outlier of multi-variable. The landslide deformation data can be visualized through a variety of methods. From the form of expression, it is better to separate the horizontal deformation data and vertical deformation data, and it is easier for people to obtain relevant information from the figure 12 and 14. While figure 13 shows the data of North, West and Height at the same time, it cannot well show the deformation process, and only shows the results. Thank you for your comment, we will improve our manuscript in our new version.

[Figure]

**Fig. 2.** Figure 13 Surface displacement visualization.jpg

**Fig. 3.** Figure 14 3D graph on April 8th 2019.jpg

[Figure]

---

## Referee Comment (RC2) · Anonymous Referee #2 · 13 Jul 2020

[Major section] There are three major weaknesses in this manuscript so I suggest this manuscript should be rejected. First, most figures are too unclear to understand what the author's description. For example, the resolution of maps, labels, and numbers in Figs. 1, 4, 5, 6, 7, 8, 9, 10, 11, 13, and 14 are poor, so that I cannot identify the results are corresponding to text. Second, the introduction of methodolgy is not enough to explain how the results are produced. For example, Box plot, EDA, fitting method. Besides, the literature reviews should include the important results or concept to relate to this study and basic contents should be introduced, such as the distribution of layers

in Fig. 2 (Line 47: heavily deformed sandstone, siltstone, marlstone, limestone, and soil.), evidence of landslide boundary in Figs. 1 and 13, and the varation meaning of each data in Fig. 4. Third, the data and evidence are not enough to support the conclusion of rainfall-induced rotating landslide. [Minor section] It is conflict between the statements of Lines 78-90 and Lines 90-91. What is the meaning of four column diagrams in Fig. 11? If you use the same scale in North axis and West axis, it could show the azimuth directly in Fig. 12. Fig. 13: the horizontal displacement could be calculated by N and W components. Lines 158-159: "..., the average rate of 8 points is 2mm per day, ..." Is it corresponding to the data of Line 155-156 "The horizontal displacement of the GPS1 point is the largest, reaching 792mm (from July 14, 2017 to Apirl 8, 2019, 633 days)". 792mm/633days= 1.25mm per day(Largest?)

---

## Referee Comment (RC3) · Anonymous Referee #3 · 13 Jul 2020

The authors have presented a landslide monitoring from GPS measurements. However, it is hard to read symbols from the current PDF figure quality. The GPS locations are only can be recognized from Fig. 14 1. The GPS specification is not provided, such as GPS, Beidou, or others, and single or dual-frequency. 2. The reference GPS8 is outside the landslide, but how far away will affect the accuracy of GPS solutions. 3. The scatter plot in Fig. 11 is not sufficient to support landslide behavior. Probably horizontal displacement can explain more sliding conditions. 4. Since GPS8 is outside the landslide area, the authors should explain why it is moving in Fig. 12.

---

## Short Comment (SC1) · 15 Jul 2020

The authors have incorporated the main changes suggested in the revision of the paper. So, I recommend to accept publication.
* * *

---

## Author Comment (AC3) · 15 Jul 2020

[Figure]

Figure 1 Location of the studied landslide site

Figure 2 Positions of the monitoring instruments

[Figure]

Figure 3 Monitoring instruments on the landslide

[Figure]

(a) Soil water content, pore water pressure and rainfall

(b) Inclinometer and rainfall

(c) GPS horizontal displacement and rainfall

(d) GPS vertical displacement and rainfall

Figure 4 Relationship between rainfall and the other monitoring data

[Figure]

Figure 5 Box plots of GPS

[Figure]

Figure 6 Vertical displacement and its fitting

[Figure]

Figure 7 Vertical displacement and its fitting

[Figure]

Figure 8 Scatter plots of GPS1 and GPS2

[Figure]

| #obs | R^2 | const a | std-err a | t-stat a | p-value a | slope b | std-err b | t-stat b | p-value b |
|------|-----|---------|-----------|----------|-----------|---------|-----------|----------|-----------|
| 608 | 0.900 | 0 | 0.013 | 0 | 1 | 0.949 | 0.013 | 74.031 | 0 |

(a)Standardized data of GPS1

| #obs | R^2 | const a | std-err a | t-stat a | p-value a | slope b | std-err b | t-stat b | p-value b |
|------|-----|---------|-----------|----------|-----------|---------|-----------|----------|-----------|
| 608 | 0.900 | 0 | 0.013 | 0 | 1 | 0.949 | 0.013 | 74.031 | 0 |
| 8 | 0.601 | -11.873 | 4.750 | -2.500 | 0.047 | 10.659 | 3.549 | 3.004 | 0.024 |
| 600 | 0.911 | -0.015 | 0.012 | -1.301 | 0.194 | 0.928 | 0.012 | 76.324 | 0 |

Chow test for sel/unsel regression subsets: distrib-F(2,649), ratio-68.7730, p-val-0

(b) Brushing the scatter plot of GPS1

Figure 9 Analysis data of GPS1

| R^2 | const a | std-err a | t-stat a | p-value a | slope b | std-err b | t-stat b | p-value | s | R^2 | const a | std-err a | t-stat a | p-value a | slope b | std-err b | t-stat b |
|-----|---------|-----------|----------|-----------|---------|-----------|----------|---------|---|-----|---------|-----------|----------|-----------|---------|-----------|----------|
| 0.970 | 2.909 | 0.544 | 5.343 | 0.000 | -0.197 | 0.001 | -139.769 | 0 | | 0.949 | -5.095 | 0.424 | -12.016 | 0.000 | -0.117 | 0.001 | -106.563 |

| R^2 | const a | std-err a | t-stat a | p-value a | slope b | std-err b | t-stat b | p-valu | s | R^2 | const a | std-err a | t-stat a | p-value a | slope b | std-err b | t-stat b |
|-----|---------|-----------|----------|-----------|---------|-----------|----------|--------|---|-----|---------|-----------|----------|-----------|---------|-----------|----------|
| 0.939 | 4.346 | 0.621 | 6.999 | 0.000 | -0.153 | 0.002 | -96.407 | 0 | | 0.941 | 12.684 | 0.607 | 20.905 | 0 | -0.159 | 0.002 | -95.584 |

Figure 10 Scatter plots of GPS3 GPS4 GPS5 and GPS7

[Figure]

Figure 11 Scatter plot matrix between GPS1 GPS2 GPS3 and GPS4

Figure 12 GPS displacement from July 14th 2017 to May 1st 2019

[Figure]

Figure 13 Surface displacement visualization

[Figure]

Figure 14 3D graph on April 8th 2019

[Figure]

Figure 15 Zhutoushan landslide type based on Varnes 1978

---

## Author Comment (AC4) · 15 Jul 2020

We appreciate your comments, which prompted us to improve the manuscript. Box plot, scatter plot, EDA and fitting method are just methods or tools to explore and analysis the landslide monitoring data. Ok, we agree with you that we will introduce these concepts in the manuscript. The data and evidence are not enough to support the conclusion of rainfall-induced rotating landslide. From figure 4 we can know that the rainfall has caused the displacements in horizontal, vertical and underground inclination, and according to figure 12, we can obtain the information of the displacement of

the landslide: displacement quantity and direction. From figure 14, we can know the displacement of monitoring points that locate in different position of landslide in vertical direction. Combined with these figures, we can see that zhutoushan landslide is a rotational landslide. The data and evidence support the conclusion of rainfall-induced rotating landslide. Yes, the horizontal displacement could be calculated by N and W components, the azimuth direction is the same. Azi=arctan($\Delta Y/\Delta X$), we can know that the displacement quantity and direction from the Cartier coordinate system, so the azimuth need not to be calculated. The displacement quantity and displacement rate are different concepts. So we should explain it in our new version, thanks for your comments.

Fig. 1. Figure 12 GPS displacement from July 14th 2017 to May 1st 2019.jpg

[Figure]

**Fig. 2.** Figure 13 Surface displacement visualization.jpg

[Figure]

**Fig. 3.** Figure 14 3D graph on April 8th 2019.jpg

---

## Author Comment (AC5) · 15 Jul 2020

Many thanks for your positive recommend and valuable time, we will try our best to improve and enhance our manuscript according to referee's comments.

---

## Author Comment (AC6) · 15 Jul 2020

The authors are grateful to you, who offered many constructive suggestions to enhance the manuscript. We will provide the GPS specification in our new version, the GPS receivers with dual-frequency can process the data of combined positioning system which is GPS, GALILEO, GLONASS and BeiDou. The accuracy of GPS solutions is related with the quality of satellite signal, observation time, data processing model, surrounding environment and so on. Scatter plot can express the relationship which is positive correlation or negative correlation, because the trend of vertical deformation

of monitoring points which located in different positions is different. It is true that GPS8 is outside the landslide area, we will explain the reason why it is moving in figure 12 in our new version. Thanks for your comments.

––––––––––––––––––––––––––––––

---

## Referee Comment (RC4) · Anonymous Referee #2 · 16 Jul 2020

Thanks for your clear figures. "Horizontal displacement" means resultant vector of N and W components, and it can help us to know the landslide moving direction at each GPS station, such as Fig. 12. However, the ratio of X and Y is not 1 in Fig. 12. If the ratio is 1, it could let us know the moving direction directly. Vertical displacements of 8 GPS stations display on Figs. 13 and 14. Could you combine these information in one figure (Fig. 13 with scale (mm))? Fig. 15 is a rotational landslide case "after" failure, however, Zhutoushan landslide is developing rotational landslide with creeping behavior. If you could modified Fig. 15 to corresponding to the status of Zhutoushan

landslide, it will be more reasonable. Besides, the other questions should be replied or explained in new version, especially in geological profile explanation (material, layers) and boundary of landslide (scarp? lineation?).

---

## Author Comment (AC7) · 28 Jul 2020

We are grateful to anonymous Referee #2 for the valuable comments and suggestions relevant for the improvement of the manuscript. The reviewer's comments were taken into account in the revised version of the manuscript. The ratio of X and Y should be 1 in Fig 12. this is about the aspect ratio of the figure, we will modify this ratio in our new version of the manuscript and let everyone know the moving direction clearly and directly. Fig 14 is a three dimensional graph and demonstrate the vertical displacement, it seems very intuitive and easy to understand. Fig 13 shows three elements (North,

[Figure]

West, Hight) on a planar graph. If we combine these information in one figure, this is a good ideal that we can try it in the future, however it is easy to understand that we separate them. the scale in the figure 13 is mm, I forgot to add this scale. Thanks for your hint. Fig 15 is not only just a rotational landslide showing failure, but a developing landslide with creeping behavior. In my new version of manuscript, we will add the geological profile explanation, and change the boundary of the landslide from points to lines so that it is easier to understand. Many thanks for your positive comments and valuable time to improve the manuscript.